# Incorporation of Biotechnologies into Gene Banking Strategies to Facilitate Rapid Reconstitution of Populations

**DOI:** 10.3390/ani13203169

**Published:** 2023-10-11

**Authors:** Harvey D. Blackburn, Hymerson Costa Azevedo, Phillip H. Purdy

**Affiliations:** 1USDA ARS National Animal Germplasm Program, 1111 S. Mason St., Fort Collins, CO 80521-4500, USA; 2Brazilian Agricultural Research Corporation (Embrapa), Aracaju 49025-040, SE, Brazil; hymerson.azevedo@embrapa.br

**Keywords:** gene bank, genetic diversity, assisted reproductive technology, clone, germplasm, primordial germ cell, somatic cell, sex-sorted semen

## Abstract

**Simple Summary:**

Gene banks need to respond to a variety of stakeholder needs, including rapid large-scale reconstruction of populations that may have been lost due to disease epidemics or are at risk from other causes. To date, reconstitution efforts have focused upon backcrossing schemes with semen, the use of embryos, or gonadal tissues, but there are a variety of assisted reproductive technologies that can be used that may be more effective approaches for reconstitution. For example, approaches such as in vitro fertilization combined with sex-sorted semen and primordial germ cells combined with gene-edited host chickens have not been explored in terms of the efficacy and efficiency for reconstitution. Due to the potential cost savings and biological efficiencies, it is imperative to determine how such approaches may be incorporated into gene bank collection strategies. Consequently, this manuscript describes these models and suggests gene bank collection goals for germplasm samples that are necessary to achieve success.

**Abstract:**

National animal gene banks that are responsible for conserving livestock, poultry, and aquatic genetic resources need to be capable of utilizing a broad array of cryotechnologies coupled with assisted reproductive technologies to reconstitute either specific animals or populations/breeds as needed. This capability is predicated upon having sufficient genetic diversity (usually encapsulated by number of animals in the collection), units of germplasm or tissues, and the ability to reconstitute animals. While the Food and Agriculture Organization of the United Nations (FAO 2012, 2023) developed a set of guidelines for gene banks on these matters, those guidelines do not consider applications and utilization of newer technologies (e.g., primordial germ cells, cloning from somatic cells, embryo transfer, IVF, sex-sorted semen), which can radically change how gene banks collect, store, and utilize genetic resources. This paper reviews the current status of using newer technologies, explores how gene banks might make such technologies part of their routine operations, and illustrates how combining newer assisted reproductive technologies with older approaches enables populations to be reconstituted more efficiently.

## 1. Introduction

The purpose of this manuscript is to provide perspectives on assisted reproductive technologies and encourage gene bank programs to fully exploit their utility. To accomplish this, it is important to have a proper perspective on gene bank collection goals and operations. Views and practices of conserving livestock genetic resources have been dynamic over time, as exemplified by the 1986 symposium at the 3rd World Congress on Genetics Applied to Livestock Production [1,2,3,4]. At that symposium, the discussion was focused upon maintaining in situ populations that were genetically rare or unique. Land [3] considered the in situ conservation of populations futile, principally due to their lack of economic competitiveness. During this time frame, Smith [5] showed the utility of such populations by computing future values. By the late 1990s and early 2000s, programs and strategies for conserving animal genetic resources (AnGR) were becoming established (US, Dutch, and French) [6,7,8]. While some national programs included in situ populations (e.g., Brazil), most efforts focused upon developing gene banks for storing germplasm from the primary livestock species. Also, at this time, much of the interest was centered around rare breeds of livestock, which was largely tied to their small numerical populations. However, several countries (Brazil, Canada, France, U. S.) proceeded in establishing gene banks for all their livestock breeds [8].

Initial germplasm targets for most national breed collections were generally based upon the guidelines developed by FAO [6]. In general, these guidelines suggested germplasm quantities, which vary by species, be 200% of that needed to reconstitute a breed either by backcrossing or using embryos. In addition, the target effective population size was set at 50 animals. This number was chosen because an effective population size (Ne) of 50 would result in an inbreeding rate of 1% per generation. Under the backcrossing scheme, a large proportion of the cryopreserved semen would be used in creating the first filial generation (F1). As subsequent generations of backcrossing occurred, the number of upgraded females would become smaller than the previous generation. The amount of cryopreserved semen needed to accomplish this would be relatively large. While reconstitution with embryos requires smaller collections, these collections are more expensive to acquire and the genetic complement is fixed, not allowing users to make mating decisions at the gametic level.

During the course of early collection development, the limitations of the original FAO [6] guidelines were identified, and development of a new set of cryo-guidelines was initiated in 2008 and published in 2012 [9]. In these guidelines, a new approach for reconstituting breeds was presented. This approach suggested breeds can be reconstituted through backcrossing with smaller quantities of semen. In addition, this set of guidelines put forward the collection of oocytes, spermatogonia, and somatic cells as additional tissues that can be used in reconstituting breeds. At the time of writing the second version of the guidelines, primordial germ cells were also identified as a tissue type that may be useful with more development. Recently, FAO [10] released a new set of cryo-conservation guidelines, and although the new guidelines did not recommend specific quantities of germplasm or tissue to collect, they did discuss the utilization of primordial germ cells (PGCs) in chickens in conjunction with the use of a gene-edited host.

Progress made in developing cryopreserved collections in gene banks has been substantial in recent years, especially in Brazil and the United States [11,12]. However, the primary emphasis has been to collect samples from animals that represent the genetic diversity contained within a breed and could be used to reconstitute a breed with a relatively small number of animals with an Ne of 50. While this continues to be a major emphasis of gene banking livestock species, it does not consider other functions the gene bank can serve and how new reproductive technologies can be used to increase the efficiency of gene bank collections. In addition, as the livestock sector faces new challenges of climate change and increasing threat of disease, which may cause epidemics of national or multinational proportions, gene banks may be called upon to respond by reconstituting populations at a scale meaningful to the industry at large. Our purpose, therefore, is to review technologies of interest and explore how gene banks might use them to respond at an industry-wide scale during times of crisis.

## 2. Background

### 2.1. Assisted Reproductive Technologies (ARTs)

Traditionally, gene banks have embraced the model that, as is often the case with cattle, one straw of semen is used for one artificial insemination (AI) of a cow and should result in the production of one calf. This model has also been applied to most other mammalian species. However, that same model is slightly different when applied to aquatic, avian, or insect species. Here, it is understood that insemination of one or two semen straws can be used for a pool of eggs (aquaculture) or to maintain the production of fertile eggs for days or weeks (e.g., chicken), months (e.g., turkey), or years (e.g., honeybee).

However, as ARTs are improved, or new ones are created, a gene bank needs to consider how their application will impact germplasm collection goals for a species. In addition, when considering germplasm collection goals, it is imperative to ensure that although an ART system may have become more efficient and require the collection of fewer germplasm samples per animal to produce the same or greater numbers of progeny, that progeny must represent the full genetic diversity of a line or population. The point being that increased efficiencies with ARTs will never replace the need for a gene bank to perpetually analyze and monitor the genetic diversity of a line, breed, or species for in situ and ex situ populations.

### 2.2. ARTs for Chickens

Semen cryo-preservation and AI can be successful [13,14,15], but the techniques produce highly variable results [16]. Furthermore, the recreation of a line (research, commercial, or heritage) using frozen semen and AI may require at least four backcrosses, depending on the mating scheme and the genetic attributes of the line (e.g., single gene mutation) [9,13].

Despite the issues, frozen rooster sperm can still provide valuable contributions to a gene bank, but it is imperative to understand the effects that will dictate success or failure when using AI with chickens. Furthermore, similar to aquatic species, each straw of rooster sperm can often be used to inseminate multiple females, usually one to five per straw, but again, this is determined by the male and will also be affected by the age of the rooster. Moreover, the method of AI (vaginal or intramagnal), will likewise impact the fertilizing potential of an insemination dose. Typically, vaginal inseminations will remain fertile longer than intramagnal inseminations because the hen can utilize her sperm host glands, crypts in her reproductive tract that hold quiescent sperm until they are needed for fertilizing ova, most efficiently with that method. That said, the report by Bacon et al., [17] demonstrated that high levels of fertility can be achieved with intramagnal AI.

Furthermore, frozen-thawed rooster sperm is of value for propagating chicken lines once progeny are produced. It can be effective for backcrossing to regenerate lines and broadening the genetic base of a population. However, because rooster sperm is homogametic and only contains the Z chromosome, repopulation strategies are genetically incomplete due to the W chromosome not being represented.

An alternative method to preserve chicken germplasm, gonad vitrification, and transplantation offers some advancement in conservation. Use of these technologies enables laboratories to collect and vitrify large quantities of gonads (>100 chicks/day) from young chicks (1 to 7 days of age). However, this methodology is difficult to scale up because it is technically challenging to perform and raises animal welfare issues for some institutions.

### 2.3. Chicken PGCs

The use of PGCs is emerging as a viable and effective means to preserve poultry germplasm. Originally, transmission rates using freshly isolated PGCs were quite low (approximately 10–15%) [18,19], but developments in culture methodologies to increase PGC numbers before and after cryo-preservation or vitrification have enabled greater success (>50%) [20,21,22,23,24]. Consequently, preserving sufficient samples for repopulation of a breed or line should be affordable. However, activities related to the repopulation itself may represent a significant expense [25], especially if the reconstituted animals and their offspring are chimeras. 

Creating chimeric chickens, or any other species, for use in repopulation schemes is problematic because the first generation (chimera) only contains half the desired genome of the population being reconstituted, and substantial back breeding involving multiple generations is required to increase the genetic purity of the offspring to an acceptable level. However, with chickens, this problem can be overcome if sufficient PGCs are used for transplantation into a host embryo and the gonad of the host is devoid of endogenous PGCs (sterile). Thus, that host embryo will develop into a chicken in which the PGCs that populate its ovary are exclusively transplanted from the donor. Two sterile host models have been generated to date. The first model used TALEN-mediated gene targeting to silence DEAD-Box Helicase 4 (*DDX4*), which is important for the development of germ cells; however, female PGCs did not readily proliferate in culture after disruption of *DDX4* [24,26]. Recently, a second model was developed, an inducible sterile host (ISH) produced by inserting truncated human caspase-9 (iCaspase9) in the last exon of the Azoospermia Like (*DAZL*) gene, which is a known determinant of avian germ cells. The ISH model has been successful for the transplantation of male and female PGCs, as well as those from different breeds. When an iCaspase9 rooster is produced, his sperm can be used to produce sterile female embryos for transplantation [26]. At sexual maturity, the recipient hen can be inseminated with sperm of the same breed or line as the transplanted PGCs, using either frozen-thawed samples and artificial insemination or a sire-dam surrogate mating system where both the rooster and hen are surrogate hosts [24], and the progeny will be fertile and capable of progenerating the line. This method [24,26,27] is a game-changing approach for biobanking chicken genetic resources as the cryopreserved population can be reconstituted in one generation with the genetic unity.

### 2.4. Fibroblasts

Cloning is rapidly becoming a more viable ART for AnGR programs. While the success rates are still low, and in some instances the costs may be prohibitive, commercial entities and laboratories are using this ART routinely. Cloning research is only being explored on a limited scale due to the high costs associated with its procedure, resulting from its low efficiency [28], but it is important to remember the value and limitations of cloning technologies. Cloned pigs produced by somatic cell nuclear transfer (SCNT) derived from superior boars produced semen with normal quality and exhibited similar reproductive performance as the donor boars, whose progeny showed greater growth performance than those derived from non-cloned pigs. Then again, the developmental rate of the cloned pig embryos, representing cloning efficiency, were between 0.4 to 2.4% [29,30,31], which seems low, but significant numbers of clones can be generated with each round of SCNT.

The simplicity of collecting tissue biopsies by scalpel, ear notch, or tissue sampling unit (AllFlex)—followed by digestion, culture of the fibroblasts, and cryo-preservation—lends itself to most established laboratories, and the technologies can be taught in a very short time. Moreover, the technologies are affordable, and a significant number of samples can be collected and/or cultured simultaneously.

In addition, fibroblasts can be used in the same modeling scenarios as PGCs. While the PGC transplantation and mammalian cloning process are distinctly different, fibroblasts, like PGCs, can be repeatedly frozen, cultured, utilized, and an aliquot of the culture frozen again. It is anticipated that fibroblasts, like most other diploid mammalian cells, can undergo 15 to 50 passages when in culture without reaching replicative senescence or immortality [32,33]. Like the offspring that are derived from PGC technologies, a clone becomes a new source of fibroblasts that can be collected, preserved, cultured, and utilized repeatedly, as described previously, in addition to having the live animal in a production setting for reestablishment of any population.

Again, like the PGCs, because of the utility of a sample when used appropriately, the size and diversity of the collection will never be diminished. The critical factor will be to ensure that once a sample is intended to be used for repopulation activities, it is also utilized to replenish itself and maintain the integrity of the collection.

### 2.5. Sex-Sorted Semen

Sex-sorted semen, particularly in the dairy industry, is primarily used to produce more genetically valuable replacement females. The fertility of this type of semen in artificial insemination has improved over time, but, beyond the direct costs for its production compared to conventional semen [34,35], the primary cost of sex-sorted semen remains the indirect costs that are associated with lower conception rates. High-speed sorting procedures for sex-sorted semen production may negatively affect the sorted sperm and result in lower fertility [36]. The issues underlying lower fertility are reduced motility, acrosomal integrity, and alterations in flagella substructures, among others [37,38].

All these effects lead to sex-sorted and cryopreserved sperm having a narrow window of viability in the female reproductive tract. Therefore, they need to be available close to the site of fertilization to achieve fertility results similar to those obtained with conventionally processed semen [37,39,40,41]. However, this strategy has not been fully successful since only about two-thirds of the fertility decline of sex-sorted semen in AI (8.6%) is due to the low dose, and a third (5.0%) is due to the sorting process itself [42,43]. Consequently, to improve the efficacy of sex-sorted semen, or at least achieve an increase in the fertility rates with these sperm, it may be advantageous to combine this technology with other ARTs.

### 2.6. Sex-Sorted Semen in Combination with Other Biotechnologies

The combination of biotechnologies can enable or enhance the use of sex-sorted semen as a multiplier of genetics when used to produce embryos in vivo (multiple ovulation and embryo transfer, i.e., MOET), with in vitro fertilization (IVF), or with injection of a single spermatozoon (ICSI). These techniques have a significant advantage from the outset—they require a much smaller or even negligible amount of sperm, as they are placed alongside or inside eggs [44].

MOET embryos produced from sex-sorted semen can be frozen for preservation in genetic banks or transferred fresh as a tool for population recovery from conserved germplasm. There have been reports of a possible delay in the development of embryos produced from sex-sorted semen, but this finding has not been substantiated [45,46]. Despite these results, the use of sex-sorted semen combined with MOET is considered advantageous [46].

One very appealing attribute of using sorted sperm for in vitro production (IVP) is that considerably fewer sperm are needed for IVF, even fewer than those required for MOET. Additionally, the use of sex-sorted semen to produce IVP embryos is considered an effective way to overcome the skewed male gender ratio typically observed for IVP embryos produced with non-sorted semen [47]. Like MOET, the results of IVF with sex-sorted semen have also been inferior to those with conventional semen. Such results have been attributed to zygote dysmorphisms during the first cleavage, characterized by an increased incidence of reverse cleavage (blastomere fusion after cleavage or incomplete separation of blastomeres) and direct cleavage (cleavage of one blastomere into three, instead of the expected two) and, as reported in MOET, by delays in embryo growth, which may consequently impair the viability of embryos developing to the blastocyst stage [48,49]. Approaches that remove the sub-fertile sperm cells or, alternatively, choose only the most competent cells used to fertilize the eggs from sex-sorted semen samples, such as Percoll and swim-up methods, seem to be one way to improve the quantity and quality of IVF embryos with this type of germplasm [43,50,51].

### 2.7. Sex-Sorted Semen Performance among Species

Sex-sorted semen has been demonstrated to work well in the bovine, although at lower rates of fertility [35,36,44,52]. Ram and buck semen appear to hold up well under the physical stresses of the sex-sorting process [53], and, upon use, sex-sorted semen yielded similar fertility levels [54] as unsorted semen. A key attribute of using sex-sorted semen in small ruminants is to combine the approach with laparoscopic AI, which may alleviate some of the stresses associated with sex-sorting characterized by the impairments to sperm structure and function inherent to the technique [37,38] and avoiding transcervical AI [43,55].

## 3. Modeling Reconstitution Activities

### 3.1. Utilization of New Biotechnology for Gene Banking

In the following case studies, we explored how gene banks might use various technologies to develop genetic resource collections, and we explored how populations might be reconstituted when using these techniques, in comparison to the original backcrossing approaches. The array of technologies now available suggests gene bank managers may want to change their approach for conserving various species and breeds, including storing increased numbers of germplasm/tissue for national needs and the use of technologies that increase the ease by which the collection can be used. In addition, with the international threats of diseases (e.g., African Swine Flu—ASF) and climate change, there is a need for gene banks to consider reconstituting populations on a much larger scale that can more easily allow the industry to rebound from extreme and potentially catastrophic events.

Gene banks were originally established to reconstitute rare breeds of livestock or add genetic variability to such populations, but, according to the Domestic Animal Diversity Information System (DAD-IS) of FAO [56], only a small proportion of local breeds (9.5%) have genetic material conserved, and only 3.6% of breeds have sufficient samples for population reconstitution. (See FAO [56] for descriptions of gene banking activities by country.) Despite this, collection development has occurred within several countries [11], especially since 2018 [56]. Furthermore, experiences in the U. S. have shown gene banks have greater functionality than originally envisioned [57]. Work to date has largely been focused on using semen, blood, or, to a lesser extent, embryos; however, the efficiencies of newer biotechnologies (e.g., IVF), as demonstrated by Dechow et al. [58] in reconstituting Holstein bulls with a Y chromosome no longer in the in situ population, have revealed the utility of such technologies and the potential broader application of their purposes.

### 3.2. Rebuilding the National Herd of Pigs for Commercially Important Pigs

In recent years, the threat of an outbreak of ASF has increased. As seen in countries where outbreaks have occurred, the results can be quite severe. For example, it is estimated China lost 43 million pigs, approximately 9.6% of the national herd, due to ASF in 2015 [59]. With such losses in mind, scenarios were explored that assessed the quantities of germplasm needed to reconstitute a commercially important breed of pigs. If a similar event occurred in the U. S. and Brazil, approximately 600,000 and 210,000 pigs would be lost from the national herd of approximately 6 and 2.1 million sows, respectively.

A target quantity of 1000 sows/gilts was set for this example. The targeted population would have a genetic composition of 93.7% of the population being reconstituted (Table 1). By regenerating a population of 1000 sows, the number of animals within the reconstituted breed can be rapidly expanded to a commercial scale. It is assumed that 20 or 30 sows from a different population would be available for use and mated with a conception rate of 50 or 60%, an average litter size of nine, two matings per year, and a mortality rate of 27.8% from birth to reproductive age; all of these parameters are based upon U. S. industry averages. Table 1 indicates that the number of units of semen needed will surpass 35,000 straws (0.5 mL volume), which exceeds the FAO guidelines [9] that computed that 4200 to 9000 doses are necessary. With such an increase, it becomes evident that technologies that can reduce the quantities of germplasm necessary for reconstitution are critically needed. Once a population with 93.7% is formed, it can be used as a basis to build a larger population. For example, using the 1000 gilts with fresh semen from contemporary boars, pregnancy rates and litter sizes will likely increase (e.g., 95% and 10, respectively), so that, in another two generations, between 150,000 and 180,000 gilts will have been generated and put into production. Depending upon need, the boars from the breeding up program can be used on the 87.5% sows to generate even more pigs that have more than 90% of the reconstituted population’s genetics.

### 3.3. Rebuilding Cattle Populations

Cattle, and ruminants in general, with their lower prolificacy rates and long growth cycle, take longer to reconstitute. Using semen and backcrossing, it can take from 8 to 10 years to reconstitute the original population [9]. In addition, the number of females needed to implement a backcrossing program is large given the low prolificacy levels of these species. FAO [9] suggests 250 females are needed to achieve a reconstituted population with a Ne of 55. While embryos can reduce the time needed for reconstitution, they are costly to collect and cryopreserve, and they do not afford a gene bank’s collection any genetic flexibility, as do semen samples from a wide range of males. Additionally, a bank of embryos would have more limitations in scaling up large, reconstituted populations if needed.

We explored using IVF and sex-sorted semen to reduce the amount of semen needed in the reconstitution process and whether the initial number of cows used in the mating plan could be reduced. Using IVF with conventional semen can reduce the amount of semen used in the reconstitution process, and, while it results in a higher proportion of heifers produced for use with 93% of the targeted genome, the proportion of reconstituted to initial cow number is less than unity (Table 2).

The combination of sex-sorted semen and IVF, previously discussed, opens new possibilities for reconstituting populations at a larger scale. Table 2 demonstrates how both technologies, when used together, can increase prolificacy in the bovine. Using these technologies in tandem can create a path for scaling up to large population sizes. While the approach can increase reconstituted population sizes in relatively short order, other issues may have to be overcome, such as having a sufficient number of recipient cows and/or the ability to process and implant the newly created embryos in a timely manner.

### 3.4. Chicken PGC Use

Combining harvested and cryopreserved PGCs with iCaspase9 KO chickens provides new approaches for reconstituting chicken lines and overcoming previous issues with low semen fertility levels, absence of the W chromosome, and surgical manipulation when transplanting ovaries or testes. Based upon results from the McGrew laboratory at The Roslin Institute, University of Edinburgh [24,27,60,61], which used two types of KO host hens, we anticipate that if we start with 100 donor eggs, we should be able to achieve the following:

About 64 of the eggs/embryos will be of an acceptable quality. Once the PGCs are harvested and placed into culture, 50% of the PGC samples (32 embryos) will reach a population of 50,000 cells within 4 weeks and can be cryopreserved.If pooled PGCs (multiple donors per transfer) from each of the 32 donors are transferred to iCaspase9 hosts (32 donors × 50,000 PGCs = 1,600,000 PGCs. 1,600,000 ÷ 10,000 PGCs/host = 160 hosts), 65% of the hosts, or 104 (0.65 × 160) of the hosts/eggs, will hatch.Of those 104 hosts/eggs that hatch, 80% or 83 of the chickens will be fertile.Assuming a 50/50 sex ratio, 41 or 42 of the 83 fertile chickens will be female and can be used for AI at sexual maturity.Once inseminated, the hens in the referenced research will lay 5.3 eggs per week; thus, from these calculations, the hens can produce a total of 217 to 223 donor-derived chicks per week.

Again, based upon our experiences and those at the Roslin Institute, we provide in Table 3 target numbers for reconstituting either heritage breeds or research lines and a commercial population. Using the FAO [9] target for an effective population size of 50 suggests a larger number of donors is needed than would be necessary for some purposes. To reconstitute commercial populations, it may be desirable to extract PGCs from an increased number of donors to both broaden the genetic base and have sufficient PGCs to reconstitute an industry population with greater speed. Here, we used 150 birds (25 male and 125 female) to speed the reconstitution process. With these numbers, it would be possible to build a population with over 1000 hens after the first month of hatching.

The advantage of using the KO sterile host chickens and transplanted PGCs is that there is no backcrossing to create birds with the targeted genetics because the offspring are the product of parental germline transmission, thus saving time when regenerating research, breeds, or commercial chicken lines. In real time, this means that, for production scale purposes, 38 weeks following transplantation of PGCs, chicks are hatched that are the product of the germline transmission of the PGCs.

The speed with which populations can be reconstituted when using PGCs is not the only benefit derived from this technology. Early stages of use of the technology struggled to achieve strongly positive results because the number of PGCs collected, even when pooled from multiple donors, was often low and insufficient to populate a host gonad. The development of serum- and feeder-free culture systems [20,24,26] allows PGCs to be cultured for months and maintain adequate doubling times and number of days to confluence for each passage. Consequently, it can be envisioned that once a sample of germplasm, regardless of type (blastoderm, blood, or gonad), is collected, it can be either immediately frozen for post-thaw culture/use or it can be digested, if necessary, placed in culture, and, when sufficient PGCs are present (>250,000), it can be frozen for future use. When needed, it can be thawed and placed in culture. However, it will be most valuable to not inject all of the cells into the host and instead culture the cells so that the genetics from the animal/line collected at that time are still in the collection and can be used repeatedly. Likewise, they can be collected, frozen, cultured, and refrozen and cultured time and again, creating a sample with significantly more utility than a single use, especially considering that PGCs have been known to remain healthy in culture for months to years.

### 3.5. Fibroblast Tissues for Cloning

Fibroblast tissue harvested from ear notches provides an inexpensive approach to conserving genetic resources. Fibroblast samples would be used to create clones of the animal sampled. Therefore, enough animals need to be sampled to obtain an effective population size of 50. Creating cloned animals from the tissue harvested can be relatively expensive, and one commercial laboratory quoted a price of $10,000 per clone (personal communication). However, such fees might potentially be reduced if cloning was performed at a public institution or if market forces work toward lowering the cost. For ruminant species, the cost of cloning for reconstituting populations may be similar in magnitude to using semen in a backcrossing program that takes 8 to 12 years to complete.

Cloning may have more utility for swine populations due to the high prolificacy of the species. Large litter sizes, as shown in Table 1, and short generation interval will decrease time required to reach targeted population sizes. Table 4 illustrates that, by using 25 boars and 25 cloned sows, by the end of year three and generation 2 there would be more than 1000 sows in production for the reconstituted population. This example assumes no cryopreserved semen is used; rather, the matings are performed by the boars (AI with cooled semen) generated from clones and resulting progeny. Of course, reconstitution could be carried out using only cloned gilts and cryopreserved conventional semen or even sex-sorted semen that could, as techniques develop and become applicable, accelerate the process. If we then assume 50 gilts were the result of cloning, and cryopreserved non-sorted semen plus live matings (in generation 2 and 3) were used, an additional 899 gilts/sows would be in production by the third year.

For pigs, utilizing cloning and cryopreserved semen appears to be an important option to consider when reconstituting populations. However, the decision point on whether to use cloning would be based upon time limitations and whether the entity reconstituting the population has the financial resources required for a particular approach. If cloning is chosen as the approach for reconstituting populations, it is suggested semen from additional males be cryopreserved to provide additional genetic variability to the reconstituted populations.

## 4. Discussion

Our review of various techniques to secure AnGR by gene banks suggests that there are a range of technologies that have to date been underutilized in gene banking of genetic resources. Industry-based experiences with sex-sorted semen, for example, have become routine in the bovine and can be executed with relatively high levels of repeatability with that and other mammalian species. While still early in development, capturing and conserving chicken PGCs to be used in conjunction with a gene-edited hen holds promise and provides a path forward to better conserve poultry populations. Fibroblasts harvested from swine or other mammalian species’ ear notches, followed by cloning, also provides an avenue to more easily capture genetic variability in the gene bank and for more rapid reconstitution of targeted populations.

Gene bank managers will have to assess the incorporation of these technologies into their respective programs. It is likely that gene banks will choose to use these approaches to augment their collections of semen, embryos, and other tissues. Taking such an approach broadens the array of options available to gene banks to execute their conservation mission. Therefore, validation of the newer approaches should be undertaken by gene banks. Additionally, gene banks will have to assess their infrastructure to determine whether they have physical capacities and human resources necessary to execute the collection, cryo-preservation, storage, and reviving of the cryopreserved cells. For example, in our experience with harvesting PGCs in chickens, it was determined that such an effort could be incorporated into the program with little additional expense and relatively short training times. However, sex-sorting semen requires substantial investment in equipment or a contract with a company that owns the equipment necessary for sorting.

Complete validation of technologies used to reconstitute populations of interest has been less than comprehensive. What complicates this further is that few examples are available in the published literature to validate gene bank reconstitution activities for cattle [9,58] (p. 76), pigs [9] (pp. 77–78), or other species. In part, this is due to the confidence in a particular technique based upon accumulated research reports and industry experiences. At this time, the practical application of using fibroblasts for cloning is increasing in the private sector, which can serve as validation for gene banks. Furthermore, experiences with the combination of PGCs and gene editing in chickens are increasing among various public and private sector entities.

During the reconstitution process, the stores of germplasm will be depleted. Gene banks may choose to not use all the stored germplasm of a particular population in the reconstitution process. In addition, once the reconstitution process has been completed, the gene bank could collect new samples from the in situ population. This would be essential for the mammalian species. That said, options also exist to generate surplus cells under culture from either fibroblast or PGC origins. This suggests a near inexhaustible supply of cells for future use.

## 5. Conclusions

To date, gene banks have largely focused on acquiring stores of germplasm that would enable them to reconstitute relatively small populations of animals. However, in an era of increased disease threats, wars and trade disputes, significant climate change, relatively thin profit margins, and errant breeding decisions by the industry, it may be incumbent upon gene banks to support national livestock, poultry, and aquaculture sectors at much larger scales than originally envisioned. Coupled with this expanded role is the need to use the full array of biotechnologies that can speed reconstitution. Therefore, we recommend gene bank managers bolster collections to 1. Broaden collections to include tissues used to employ new biotechnologies (e.g., PGCs, ear notches); 2. Increase the number of samples and animals for important breed collections, enabling reconstitution of populations capable of facilitating a national recovery effort in the event of a significant epidemic or other catastrophic losses; 3. Convey the importance of using gene bank collections to various policy makers so there is government-wide recognition of the collection’s utility.

## Figures and Tables

**Table 1 animals-13-03169-t001:** Modeling the repopulation of pigs to acquire 1000 sows/gilts with 93.7% of the targeted genetic composition.

Scenario	Initial Sows	93.7% Sows ^a^	Total Matings	Total 0.5 mL Straws	Years	Parameters
1	20	1091	1254	37,620	3.5	27.8% = Mortality Conception = 50% 1 mating at 87.5% ^b^ Starting Sows = 0% ^b^
2	20	1151	1358	40,740	3.5	27.8% = Mortality Conception = 60% 1 mating @ 75% ^b^ 1 mating @ 87.5% ^b^ Starting Sows = 0% ^b^
3	30	1023	884	26,520	2.5	27.8% = Mortality Conception = 50% Starting Sows = 50% ^b^

^a^ Refers to the genetic composition needed to reestablish a population by industry standards. ^b^ Percent of target population genetic composition.

**Table 2 animals-13-03169-t002:** Modeling the repopulation of cattle to acquire heifers with 93.5% of their targeted genetic composition.

	Standard AI	IVF + Non-Sorted Semen	IVF + Non-Sorted Semen	IVF + Sex-Sorted Semen	IVF + Sex-Sorted Semen
Founder cows	250	80	200	50	200
93.5% heifers ^a^		27	68	180	719
Units of semen	1410	112	280	155	624
Ne	55	50	50	50	50
Number of bulls	25	23	15	13	13
Semen units/bull	57	5	19	11.5	49

^a^ Refers to the genetic composition needed to reestablish a population by industry standards.

**Table 3 animals-13-03169-t003:** Utilizing PGCs for reconstituting chicken populations.

Contribution	Research or Heritage Breed	Commercial Population
PGC donors (male:female)	50 (1:1)	150 (1:5)
Hosts generated (0.65) ^a^	97	231
Fertile host eggs (0.8) ^b^	63	150
Fertile chickens (male:female)	50 (1:1)	120 (1:5)
Eggs produced in 4 weeks ^c^	530	2120
Ne	50	83

^a^ Estimated that 65% of the host eggs generated (*n* = 231 in this example) are fertile. ^b^ Estimated that 80% of the 231 eggs laid are fertile and result in the production of 20 roosters and 100 hens. ^c^ Based on an estimate of 5.3 eggs laid per week per hen, 4 weeks, and 100 hens.

**Table 4 animals-13-03169-t004:** Models for the use of fibroblasts and cloning for pig population rejuvenation and the resulting quantity of gilts/sows in production.

Year	Generation			Total
	1	2	3	
0	25 (50)	25 (50)	25 (50)	25 (50)
1		147 (295)	147 (295)	147 (295)
2			874 (1748)	874 (1748)
				1046 (1945)

Mated with a conception rate of 60%, an average litter size of nine, two matings per year, and a mortality rate of 27.8% from birth to reproductive age. Numbers in parenthesis are when 50 boars and 50 sows are cloned, and all other parameters remain the same.

## Data Availability

Data is contained within the article or the data presented in this study are available in the cited works.

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
