# Peer review of "Incorporation of Biotechnologies into Gene Banking Strategies to Facilitate Rapid Reconstitution of Populations"

_animals, 2023, doi:10.3390/ani13203169_

Round 1

Reviewer 1 Report

Title: Incorporation of Biotechnologies into Gene Banking Strategies to Facilitate Rapid Reconstitution of Populations

Authors: Harvey D. Blackburn, Hymerson Costa Azevedo, and Phillip H. Purdy

Manuscript ID: ANIMALS-D-21-xx

Review:

I have comprehensively reviewed the manuscript "Incorporation of Biotechnologies into Gene Banking Strategies to Facilitate Rapid Reconstitution of Populations" by Blackburn, Azevedo, and Purdy. Overall, the paper contributes to gene banking and conserving animal genetic resources. The authors provide a detailed exploration of various biotechnologies that can enhance the efficiency and effectiveness of gene banking practices. Here are my specific comments and recommendations:

Abstract:

The abstract provides a clear and concise summary of the paper's objectives and key findings. It sets the stage for the rest of the article by highlighting the importance of incorporating biotechnologies into gene banking strategies. However, it could be enhanced by briefly mentioning the specific biotechnologies discussed in the paper (e.g., PGCs, sexed semen, cloning) to provide readers with a more precise roadmap of what to expect in the article.

Introduction:

The introduction offers historical context and evolution in the field of gene banking, which helps understand the background. However, it is lengthy and could benefit from a more concise presentation. Additionally, while the introduction introduces the concept of incorporating biotechnologies, it might be more engaging if it explicitly states the research gap or problem this paper aims to address.

Background:

The background section provides a comprehensive overview of various assisted reproductive technologies (ARTs) and their applications in gene banking. However, it is dense and may benefit from breaking down the information into smaller, more digestible sections or subsections. Additionally, the team could be improved by including more recent references to reflect the field's current state.

Discussion:

The discussion section effectively highlights the underutilization of specific biotechnologies in gene banking and advocates for their incorporation into regular gene banking practices. It appropriately addresses practical considerations and emphasizes the need for validation. However, it could benefit from a more detailed exploration of potential challenges and limitations associated with each technology discussed.

Conclusions:

The conclusions aptly summarize the paper's main findings and reiterate the need for gene banks to adapt and adopt new biotechnologies. However, providing specific recommendations or action points for gene bank managers, policymakers, and researchers interested in implementing these technologies might be valuable. This would make the conclusions more actionable.

General Comments:

1. The paper could benefit from improved organization and flow, particularly in the background section, which is densely packed with information. Breaking down the content into smaller units or using subsections would enhance readability.

2. Throughout the manuscript, providing more recent references to reflect the latest advancements and developments in gene banking and assisted reproductive technologies would be beneficial.

3. The paper could benefit from a more explicit statement of the research problem or research questions at the beginning of the introduction to provide readers with a clear understanding of the paper's objectives.

4. It might be helpful to provide real-world examples or case studies of gene banks that have successfully incorporated these biotechnologies, demonstrating their practical applications and benefits.

In summary, the manuscript presents valuable insights into the potential of biotechnologies to improve gene banking practices. With some revisions to improve organization, clarity, and engagement, this paper can contribute significantly to the field, including an English review.

The article needs an English review, the Word file includes some English corrections.

Author Response

Abstract:

The abstract provides a clear and concise summary of the paper's objectives and key findings. It sets the stage for the rest of the article by highlighting the importance of incorporating biotechnologies into gene banking strategies. However, it could be enhanced by briefly mentioning the specific biotechnologies discussed in the paper (e.g., PGCs, sexed semen, cloning) to provide readers with a more precise roadmap of what to expect in the article. 

Response: line 29 – 30 was modified to accommodate this request.

Introduction:

The introduction offers historical context and evolution in the field of gene banking, which helps understand the background. However, it is lengthy and could benefit from a more concise presentation. Additionally, while the introduction introduces the concept of incorporating biotechnologies, it might be more engaging if it explicitly states the research gap or problem this paper aims to address. 

Response: we disagree with this perspective.  The information in the Introduction provides a historical perspective on gene bank development which is necessary to understand why collections have developed in the manner that they did.  Our perspective, and the purpose of this manuscript (i.e., the research gap or problem this paper aims to address) is stated in lines 88-90.

Background:

The background section provides a comprehensive overview of various assisted reproductive technologies (ARTs) and their applications in gene banking. However, it is dense and may benefit from breaking down the information into smaller, more digestible sections or subsections. Additionally, the team could be improved by including more recent references to reflect the field's current state

Response: A similar suggestion was proposed by another reviewer as well.  To comply with both, we edited the PGC section and moved some of the information to Section 3.  In addition, we edited the section of content.  However, we disagree with the suggestion about the “recent references”.  We believe that the references chosen are appropriate because they add historical context (e.g., the Introduction) or they substantiate a point we have made.  The age of a reference does not invalidate its truth.  Furthermore, when one considers the distribution of the references, it demonstrates that we did a thorough job of surveying the literature.  The following is a listing of the year of the reference and the number of citations per year, the vast majority occurring within the 2000’s.  1961:1, 1984:1, 1986:5, 1994:1, 1998:3, 2006:2, 2007:1, 2009:5, 2010:1, 2011:2, 2012:2, 2013:2, 2014:1, 2015:3, 2016:4, 2017:5, 2018:3, 2019:5, 2020:3, 2021:4, 2022:5, 2023:2.

Discussion:

The discussion section effectively highlights the underutilization of specific biotechnologies in gene banking and advocates for their incorporation into regular gene banking practices. It appropriately addresses practical considerations and emphasizes the need for validation. However, it could benefit from a more detailed exploration of potential challenges and limitations associated with each technology discussed.

Response: thank you for the suggestion.  While we are familiar with cost benefit analyses of programs, ARTs, etc., (Silversides, Purdy, Blackburn, British Poultry Science Volume 53, Issue 5, Pages 599 – 607, October 2012), we do not believe that inclusion of that information is necessary for this commentary.  Moreover, the volume of that information will make this publication excessive and therefore it is not merited.

Conclusions:

The conclusions aptly summarize the paper's main findings and reiterate the need for gene banks to adapt and adopt new biotechnologies. However, providing specific recommendations or action points for gene bank managers, policymakers, and researchers interested in implementing these technologies might be valuable. This would make the conclusions more actionable.

Response: the Conclusion section has been modified to include this suggestion.

General Comments:

  1. The paper could benefit from improved organization and flow, particularly in the background section, which is densely packed with information. Breaking down the content into smaller units or using subsections would enhance readability.

Response: we heeded this suggestion which matched that of another reviewer.  Please see the previous response to this comment in the Background section, above.

  1. Throughout the manuscript, providing more recent references to reflect the latest advancements and developments in gene banking and assisted reproductive technologies would be beneficial.

Response: please see the previous response to this comment in the Background section, above.

  1. The paper could benefit from a more explicit statement of the research problem or research questions at the beginning of the introduction to provide readers with a clear understanding of the paper's objectives.

Response: This has been included in lines 39-41.

  1. It might be helpful to provide real-world examples or case studies of gene banks that have successfully incorporated these biotechnologies, demonstrating their practical applications and benefits.

Response: references to real-world examples have been included in lines 455-457.

Reviewer 2 Report

In this well-written review article, the authors provide insights into the integrating modern technology into Gene Banking to effectively reconstitute animal populations that are endangered or threatened due to disease epidemics or other causes. Overall, the manuscript is concise and presents important information. I enjoyed reading this manuscript and recommend publication as soon as possible. 

Author Response

Thank you for your perspectives on our manuscript.

Reviewer 3 Report

The manuscript “Incorporation of Biotechnologies into Gene Banking Strategies 2 to Facilitate Rapid Reconstitution of Populations” is a review article that addresses how gene banks can be created and how data can be improved. The authors present an interesting discussion on the topic and with an important approach for researchers in the area of biobank formation. In general, I do not see a specific point in the manuscript to be improved, being interesting for publication in its present form.

Author Response

Thank you for your consideration and perspectives.

Reviewer 4 Report

This overall well-written manuscript details a review of recent literature regarding gene banks and their role in reconstituting animal production units which may benefit from new and emerging assisted reproduction technologies. The authors explain that maintaining genetic diversity to avoid genetic bottlenecks is a goal of combining numerous genetic reconstitution techniques. Although the manuscript is readable and explained well, there are several areas that are slightly confusing with missing statements that would greatly clarify concepts for the reader that is unfamiliar with gene banking concepts, including this reviewer.  My primary concern is one of organization of the manuscript and oversimplification. For example, It is not clear why the authors chose to have a significant long section within the background section, on Chicken ARTs. It seems that they have not set up that chicken gene banking (or any other species’ needs) is a major emphasis in this paper, AND the Background section is mostly cell-based subsections and not species-specific. It seems that the species-specific solutions are discussed in the following section 3. It might be more appropriate to discuss PGCs as a separate section within the Background section, then open up more about chicken ARTs either in Section 3 or 4. Lines 162-177 seems to be more appropriately placed in Section 3 (Modeling). I suggest a major revision of the Background to more closely align chicken ARTs in the same paragraph as modeling for pigs, and cattle, then PGCs and fibroblast strategies, can then follow. This disorganization leaves the reader confused about the primary message regarding each section, as well as the overall goals. Also, the authors should discuss how banks should prepare for depletion of their specific resources when a reconstitution event is needed. Perhaps an expansion of the very short Discussion section is needed. I realize that this paper is a commentary, but as such, requires more clarification for the broad audience in the readership of Animals.

Detailed issues are discussed below: 

Introduction: what about depletion of gene banks when using for reconstitution? Should be part of strategies in subsequent sections.

Throughout the manuscript, authors need to consistently use ‘sex sorted’, ‘sex-sorted’, ‘sexed’ in numerable places. Confusing to readers.

L27: define FAO at first mention.

L143: What are the problems to overcome associated with creating chimeric chickens? Cannot assume readers will know about this, needs a better introduction to the paragraph.

L163-177: see above, this section seems out of place and better placement in section 3.

L226: Missing beginning of the sentence

L231: no verb in this sentence.

L241-243: Needs a sentence of interpretation regarding these cited numbers.  

L245-247: IVF in parentheses is misplaced in this sentence, thus doesn’t make sense.

L262: what is meant by ‘zygote dismorphisms’?

L264: Approaches such as? Needs example(s)

L274: ‘…some of the stresses…’ such as? What stresses?

L290: ‘In the following cases…’ is a critical introduction to the rest of the paper and this sentence is buried in the middle of this paragraph. This entire paragraph should be the intro paragraph for Section 3 (Modeling).

L313-317: It is decidedly not clear how or why you describe the need for >35,000 semen doses to establish 1000 sows as described in Table 1. Is there a connection between 1000 sows and re-establishing the national herd that could be lost? How does one get the estimated 600,000 to 210,000 sows replaced using 1000 sows?

Table 1: last column should list the numbers consistently for Mortality, conception, 1mating, starting sows. What is this percentage of starting sows? Indicates that better explanation is needed for this section/paragraph.

L324: ‘…8-10 years…’ for what? Needs more specific info.

L401-410: Each sentence in this paragraph should name the species intended for the statement. Eg, L408: assume ear notches is for pigs, not chickens as previous sentence indicates. 

Author Response

My primary concern is one of organization of the manuscript and oversimplification. For example, It is not clear why the authors chose to have a significant long section within the background section, on Chicken ARTs. It seems that they have not set up that chicken gene banking (or any other species’ needs) is a major emphasis in this paper, AND the Background section is mostly cell-based subsections and not species-specific.  Response: the background section and section 3 have been reorganized and should accommodate these concerns.

It seems that the species-specific solutions are discussed in the following section 3. It might be more appropriate to discuss PGCs as a separate section within the Background section, then open up more about chicken ARTs either in Section 3 or 4. Lines 162-177 seems to be more appropriately placed in Section 3 (Modeling). Response: this section has been moved to Section 3 and that section, Lines 327-379, reorganized to accommodate the change.  In addition, a new section within the Background has been created on line 138, entitled "Chicken PGCs”.

I suggest a major revision of the Background to more closely align chicken Arts in the same paragraph as modeling for pigs, and cattle, then PGCs and fibroblast strategies, can then follow. This disorganization leaves the reader confused about the primary message regarding each section, as well as the overall goals.  Response: we modified the Background section per most of your suggestions but believe the modified and current organization is most appropriate.

Also, the authors should discuss how banks should prepare for depletion of their specific resources when a reconstitution event is needed. Perhaps an expansion of the very short Discussion section is needed. I realize that this paper is a commentary, but as such, requires more clarification for the broad audience in the readership of Animals.  Response: we added additional commentary to the Discussion, lines 462-468, to address this topic.

Introduction: what about depletion of gene banks when using for reconstitution? Should be part of strategies in subsequent sections.  Response: we added additional commentary to the Discussion, lines 462-468, to address this topic.

Throughout the manuscript, authors need to consistently use ‘sex sorted’, ‘sex-sorted’, ‘sexed’ in numerable places. Confusing to readers.  Response: the manuscript was edited so that ‘sex-sorted’ or that term in a different tense, was used throughout.

L27: define FAO at first mention.  Response: corrected.

L143: What are the problems to overcome associated with creating chimeric chickens? Cannot assume readers will know about this, needs a better introduction to the paragraph.  Response: corrected per lines 144-150.

L163-177: see above, this section seems out of place and better placement in section 3.  Response: this section has been moved to Section 3 and that section, Lines 327-379, reorganized to accommodate the change.

L226: Missing beginning of the sentence  Response: corrected – the first word should have read ‘Again,’

L231: no verb in this sentence.  Response: corrected.

L241-243: Needs a sentence of interpretation regarding these cited numbers.  Response: corrected using a clarifying/transition sentence in lines 249-252.

L245-247: IVF in parentheses is misplaced in this sentence, thus doesn’t make sense.  Response: corrected in lines 255-256.

L262: what is meant by ‘zygote dysmorphisms’?  Response: clarified in lines 237-242.

L264: Approaches such as? Needs example(s) Response: clarified in lines 243-245.

L274: ‘…some of the stresses…’ such as? What stresses? Response: clarified in lines 252-253.

L290: ‘In the following cases…’ is a critical introduction to the rest of the paper and this sentence is buried in the middle of this paragraph. This entire paragraph should be the intro paragraph for Section 3 (Modeling). Response: the paragraphs in this section were moved to section 3 and reordered per your suggestions.

L313-317: It is decidedly not clear how or why you describe the need for >35,000 semen doses to establish 1000 sows as described in Table 1. Is there a connection between 1000 sows and re-establishing the national herd that could be lost? How does one get the estimated 600,000 to 210,000 sows replaced using 1000 sows?  Response: additional information has been added per this request.  Please see lines 302-308.

Table 1: last column should list the numbers consistently for Mortality, conception, 1mating, starting sows. What is this percentage of starting sows? Indicates that better explanation is needed for this section/paragraph.  Response: additional information has been added per this request.  Please see lines 302-308 for clarification of the table.  Please also see the modification to Table 1, specifically the superscript for “b”.

L324: ‘…8-10 years…’ for what? Needs more specific info.  Response: this was clarified per your suggestion in line 335.

L401-410: Each sentence in this paragraph should name the species intended for the statement. Eg, L408: assume ear notches is for pigs, not chickens as previous sentence indicates.  Response: this was clarified per your suggestion in lines 413-422.

Reviewer 5 Report

This commentary well reflects the world gene banking strategy of animals of various species. The authors focused on how to respond to the crisis that can result from an unexpected outbreak of an epidemic and the complete destruction of some species of farm animals. This is important both in epidemic and economic terms. They modelled reconstitution activities and demonstrated the need for change by incorporating new biotechnologies into routine practise in gene banks.

In general, the text is very coherent and encourages the reader to reflect. I only have a few suggestions on what could be improved.

For example:

Did the authors find information on how many countries run gene banks (and for which species)? What scale is this? This is interesting information. It would be possible to present in the form of a table the use of various biotechnologies by specific countries/gene banks and for which species genetic resources are protected.

While the question of the need to recreate certain animal populations is justified, it is not entirely clear what the purpose of recreating, for example, breeds that breeding have been stopped for some reason is. This needs to be clarified.

232-234 It is not very clear what the authors had in mind. How does the "primary cost" of sperm affect the fertilisation rate? The authors write here about the relationship between one and the other, please explain.

Author Response

Did the authors find information on how many countries run genebanks (and for which species)? What scale is this? This is interesting information. It would be possible to present in the form of a table the use of various biotechnologies by specific countries/gene banks and for which species genetic resources are protected. Response: thank you for this suggestion.  However, the list of genebank program is quite substantial and we do not feel it is appropriate considering the scope of this manuscript.  Still, to highlight your concern, and to reinforce this concept to readers, we expanded on some of the text which is highlighted in line 266.

While the question of the need to recreate certain animal populations is justified, it is not entirely clear what the purpose of recreating, for example, breeds that breeding have been stopped for some reason is. This needs to be clarified.  Response: thank your for the suggestions.  We believe this issue is described in lines 12-14 and lines 23-26 of the Summary and Abstract.

232-234 It is not very clear what the authors had in mind. How does the "primary cost" of sperm affect the fertilization rate? The authors write here about the relationship between one and the other, please explain. Response Thank you for this observation. The text really needed more clarity. We brought more elements from the articles cited to make the information clearer, as can be seen in lines 204 and 206.